# The Development of a 10-Item Ventilator-Associated Pneumonia Care Bundle in the General Intensive Care Unit of a Tertiary Hospital in Vietnam: Lessons Learned

**DOI:** 10.3390/healthcare13050443

**Published:** 2025-02-20

**Authors:** Bui Thi Huong Giang, Chieko Matsubara, Tatsuya Okamoto, Hoang Minh Hoan, Yuki Yonehiro, Duong Thi Nguyen, Yasuhiro Maehara, Keigo Sekihara, Dang Quoc Tuan, Do Van Thanh, Dao Xuan Co

**Affiliations:** 1Department of Emergency and Critical Care Medicine, Hanoi Medical University, No.1, Ton That Tung Street, Trung Tu ward, Dong Da district, Hanoi 100000, Vietnam; 2Department of Intensive Care Medicine, Bach Mai Hospital, 78 Giai Phong Road, Ha Noi 100000, Vietnam; 3Bureau of International Health Cooperation, National Center for Global Health and Medicine, 1-21-1 Toyama, Shinjuku-ku 162-8655, Tokyo, Japan; 4Department of Emergency and Critical Care Medicine, Center Hospital, National Center for Global Health and Medicine, 1-21-1 Toyama, Shinjuku-ku 162-8655, Tokyo, Japan; 5Department of Anesthesiology, Center Hospital, National Center for Global Health and Medicine, 1-21-1 Toyama, Shinjuku-ku 162-8655, Tokyo, Japan; 6Department of First Surgery, Hamamatsu University School of Medicine, 1-20-1 Handayama, Higashi-ku, Hamamatsu City 431-3192, Shizuoka, Japan; 7Department of International Corporation, Bach Mai Hospital, 78 Giai Phong Road, Ha Noi 100000, Vietnam; 8Department of Emergency and Critical Care Medicine, Faculty of Medicine, VNU University of Medicine and Pharmacy, No. 144, Xuan Thuy Street, Dich Vong Hau Ward, Cau Giay District, Hanoi 10000, Vietnam

**Keywords:** ventilator-associated pneumonia, healthcare-associated infection, care bundle, developing countries, prevention, intensive care unit

## Abstract

**Objectives and Methods:** We developed a 10-item VAP care bundle to address the high incidence of VAP in Vietnamese intensive care units (ICUs), comprising (i) hand hygiene, (ii) head elevation (gatch up 30–45°), (iii) oral care, (iv) oversedation avoidance, (v) breathing circuit management, (vi) cuff pressure control, (vii) subglottic suctioning of secretions, (viii) daily assessment for weaning and a spontaneous breath trial (SBT), (ix) early ambulation and rehabilitation, and (x) prophylaxis of peptic ulcers and deep-vein thrombosis (DVT). The VAP incidence (27.0 per 1000 mechanical ventilation days) slightly and not significantly decreased in the six months after the implementation of the care bundle. **Methods and Results:** However, the VAP incidence (11.3 per 1000 mechanical ventilation days) significantly decreased when we updated the two-item protocol with interactive communication and education (*p* < 0.001). **Conclusions:** Although the effectiveness of the interventions via protocol updates with interactive education needs further study, this intervention can make a VAP care bundle work in a resource-constrained and multidrug-resistant environment.

## 1. Introduction

Ventilator-associated pneumonia (VAP) is one of the most frequent healthcare-associated infections in intensive care units (ICUs) [1,2,3,4]. VAP is defined as healthcare-associated pneumonia occurring at least 48 h after the initiation of mechanical ventilation [1]. The mortality rate for VAP varies from 6 to 67% [5]. The disease burden of VAP is higher in low- and middle-income countries than in high-income countries [6,7]. Furthermore, VAP can result in increased hospitalization durations [8,9], higher costs of treatment and care [8,10], and frequent use of antibiotics [9,11,12]. Hence, prevention is vital for VAP management [13].

Over the past 20 years, it has been reported that a care bundle, i.e., a set of evidence-based practices [14], is effective and a core measure in the prevention of VAP [4,15], which requires respiratory management and the prevention of pathogens entering the lower respiratory tract from within and outside the tracheal tube. The Institute for Healthcare Improvement (IHI) proposed a care bundle for VAP prevention in 2004 [13]. Since then, the Intensive Care Society [16], the Japanese Society of Intensive Care Medicine [17], and other institutions have proposed VAP care bundles that selectively combine preventive measures. Subsequently, many ICUs have globally implemented the “IHI VAP care bundle”, customizing it to suit their requirements. However, as of December 2024, a review paper presenting the interventions with VAP care bundles found that 4 out of 38 studies indicated a low VAP reduction, although other studies showed a VAP reduction of 36% or more [15] and another meta-analysis paper reported that the odds ratio with 95%CI for VAP incidence did not statistically significantly reduce in 9 out of 25 studies that applied the interventions with VAP care bundles [18].

In Vietnam, VAP incidence at tertiary hospitals in Vietnam was reported at 8.7 per 1000 ventilator days in 2017 [19]; however, anecdotal evidence suggests that VAP is more prevalent and remains an urgent issue. The emergence of multidrug-resistant bacteria has further increased the importance of VAP prevention [20,21,22,23]. We developed a 10-item VAP care bundle in 2018 to address the high incidence of VAP in Vietnamese ICUs. The VAP care bundle approach showed efficacy in preventing VAP amid the COVID-19 pandemic in Vietnam [24] and Japan [25]. However, the VAP incidence was slightly and not significantly reduced during the first six months of the care bundle’s implementation. Disseminating the 10-item VAP care bundle was challenging. In 2019, we updated 2 of the 10 items of the VAP care bundle protocol with interactive education, and VAP incidence decreased. This study aimed to review how the VAP care bundle, which did not work at the time of implementation despite high compliance rates, could be made to work effectively to reduce VAP incidence via protocol updates with interactive education in a resource-poor and multidrug-resistant environment.

## 2. Methods

### 2.1. Study Site and Design

This before-and-after study was conducted at the general ICU for adult patients at Bach Mai Hospital, Hanoi, Vietnam, from September 2018 to June 2019. Bach Mai Hospital is a 3500-bed teaching hospital. We applied a before-and-after study design to directly observe changes in VAP incidence in the study population by comparing their status before and after the intervention.

### 2.2. Study Population and Sample Size

We included all patients > 18 years of age admitted to the general ICU between September 2018 and June 2019 who underwent intubated mechanical ventilation for over 48 h but did not develop VAP. This study’s exclusion criteria were as follows: patients < 18 years of age; patients who underwent intubated mechanical ventilation for <48 h and with evidence of pneumonia; patients who underwent intubated mechanical ventilation with evidence of pneumonia on admission; and patients with evidence of pneumonia before being intubated. 

The sample size was calculated as 138 [26], with an expected proportion of VAP patients of 10% (48), a 95% confidence interval, and 5% absolute precision.

### 2.3. VAP Definition and Diagnosis

We diagnosed VAP according to the diagnostic criteria of the United States Centers for Disease Control and Prevention’s National Healthcare Safety Network, defined as pneumonia occurring at least 48 h after intubation [16]. The infection control team, radiology department, and ICU physicians screened suspected VAP cases using a combination of chest X-ray images, clinical signs and symptoms, and laboratory data. Two medical doctors, including the ICU specialist, confirmed the final diagnosis. When an organism was isolated from respiratory samples, we considered it to be a “causative organism”.

### 2.4. Set-Up of 10-Item VAP Care Bundle

We developed a 10-item VAP care bundle in June 2018. The 10-item VAP care bundle, applied by Hoang et al. [24] and Sekihara et al. [25], included the following:(i)Hand hygiene [17];(ii)Head elevation (gatch up 30 to 45°): positioning the patient with the head of the bed elevated above 30°, provided there are no contraindications [14,16];(iii)Oral care: oral hygiene with 0.12% chlorhexidine following pharyngeal suction [14];(iv)Oversedation avoidance: nursing staff cease sedation after morning patient care unless contraindicated [14,16];(v)Breathing circuit management: the breathing circuit is not replaced routinely, only when it is noticeably soiled and compromised [16,17];(vi)Cuff pressure control: endotracheal tube cuff pressure is maintained within the range of 25 to 30 cm H_2_O [27,28];(vii)Subglottic suctioning of secretions: intermittent suctioning of the endotracheal tube on the cuff every 4 h or continuous suctioning is performed [16,29];(viii)Daily assessment for weaning and a spontaneous breath trial (SBT): extubation is evaluated by performing a spontaneous breathing test on the patient daily [14];(ix)Early mobilization and rehabilitation: during the day shift, the patient is permitted to sit at an angle of 70 to 90 degrees or to rise from bed at least once [30,31];(x)Prophylaxis of peptic ulcers and deep-vein thrombosis (DVT): the patient is administered medication to avert gastric and duodenal ulcers and prevent DVT as per the doctor’s orders [14].

An intensive care expert, a native Vietnamese speaker, translated the 10-item VAP care bundle from English into Vietnamese. Five medical doctors and nurses translated the Vietnamese version back into English and revised the Vietnamese 10-item VAP care bundle to improve translation accuracy. Three hospital directors examined the English and Vietnamese 10-item VAP care bundle separately to ensure that the translations were accurate and that each item’s meaning was the same in both languages [32]. Before implementation, we tested the Vietnamese 10-item VAP care bundle with 20 patients to determine whether all items were understandable.

If the item was complied with, the item was given a score of 1, and if the item was not complied with, the item was given a score of 0. The possible total score ranged from 0 to 10. We summed up the scores of the 10 items for each patient and divided the total score of the 10 items by 10 to calculate the compliance rate.

The director also decided to standardize a suction endotracheal tube that allowed for subglottic suctioning of secretions because it was scientifically proven and a supply of tubes was expected. We made a poster of the 10-item VAP care bundle and displayed it on the wards (Appendix A). We explained each care item to the staff. After two months, we started applying the VAP care bundle to all ventilated patients from September 2018 onward.

### 2.5. Updating VAP Care Bundle Protocols: ‘Oral Care’, ‘Breathing Circuit Management’, and Continuous Bedside Training

The results after six months revealed no reduction in VAP incidence despite a VAP care bundle compliance rate of 80.6%. We discussed with hospital managers again and decided to update the bundle protocols. We focused on ‘oral care’ and ‘breathing circuit management’, as we observed considerable procedural variations among nurses.

Nurses performed oral care three times per day. However, some nurses used toothpaste and rinsed the mouth with tap water because of the cost and patients’ comfort; others used a cotton swab or a toothbrush soaked in chlorhexidine; and some applied continuous suction during oral care. After discussion with managers, the use of a chlorhexidine-soaked cotton swab or a toothbrush with continuous suction was standardized without oral rinsing with water. Furthermore, the nursing protocols were revised to detail the care procedures, such as when to use hand sanitizer. An intensive care nurse specialist provided technical hands-on instruction on how to wipe the mouth (teeth, tongue, and gingiva) with a cotton ball permeated with chlorhexidine while continuously suctioning saliva and chlorhexidine during cleaning, according to the updated protocol. This updated practical guidance for oral care was made into a video-based learning tool.

Regarding the breathing circuit, single-use and multiple-use (reusable) circuits were permitted from the perspective of cost. Multiple-use circuits were to be reused after cleaning and sterilization. After discussion with the managers, single-use disposable circuits were applied for all patients. The single-use circuits enabled nurses not to change the breathing circuit frequently.

The head nurse and chief nurses conducted the training interactively at the bedside with question-and-answer sessions and practical instruction. For example, visible dirt was sometimes aspirated from the patient’s mouth when the supervisor performed oral care again after a nurse had performed oral care.

### 2.6. Data Collection and Statistical Analysis

Basic clinical data and VAP care bundle data were collected from medical records. The VAP care bundle records were collected as follows: ICU staff nurses and medical doctors recorded the 10 items of the VAP care bundle into a VAP care bundle checklist, which was included in the patients’ medical records, for each patient, three times per day (day shift, semi-night shift, and deep-night shift) via self-evaluation. The ICU head nurse and deputy head nurses monitored the 10 items repeatedly and finalized the evaluations.

The methods to evaluate compliance for each item were as follows [24]:(i)Hand hygiene: patients were observed in person during the day and via camera at night by a head nurse or a chief nurse;(ii)Head elevation (gatch up 30 to 45°): the angle of the patient’s bed joint was measured with a protractor three times a day by a staff nurse;(iii)Oral care: records were obtained from the nurse’s bedside monitoring chart three times a day by a staff nurse;(iv)Oversedation avoidance: records were obtained from the nurse’s bedside monitoring chart and medical records once a day by a staff nurse or a medical doctor;(v)Breathing circuit management: the breathing circuit was observed at the bedside three times a day by a staff nurse;(vi)Cuff pressure control: the cuff pressure was measured with a manometer at the bedside three times a day at random intervals by a staff nurse;(vii)Subglottic suctioning of secretions: records were obtained from the nurse’s bedside monitoring chart three times a day by a staff nurse;(viii)Daily assessment for weaning and a spontaneous breath trial (SBT): records were obtained from the medical records once a day by a medical doctor;(ix)Early mobilization and rehabilitation: records were obtained from the nurse’s bedside monitoring chart once a day by a staff nurse;(x)Prophylaxis of peptic ulcers and deep-vein thrombosis (DVT): records were obtained from the medical records once a day by a medical doctor.

Bundle compliance rates were calculated using database software (FileMaker Pro version 19, Claris International Inc. Cupertino, CA, USA) at least once a month. Samples of endotracheal aspirates from the lower respiratory tract were collected for bacterial cultures from all patients diagnosed with VAP.

The data were analyzed using SPSS statistics version 27 for Windows and Macintosh (IBM Corp, Armonk, NY, USA) and Stata IC 14 (Stata Corp, College Station, TX, USA). For univariate analyses, chi-square tests were performed for categorical variables, and Student’s *t*-tests were conducted for continuous variables. The VAP incidence rate (per 1000 mechanical ventilation (MV) days) was calculated by dividing the number of VAP cases by the total number of ventilator days and multiplying the resulting number by 1000 [1]. The difference in the population proportion was calculated using *z*-scores. A Kaplan–Meier analysis, a survival analysis technique that analyzes the time it takes for a certain event (such as the appearance of a disease) to occur, was performed to explore the time of appearance of VAP incidence and time of death. Deaths due to VAP and deaths due to primary disease were summed. Statistical significance was set at *p* < 0.05.

## 3. Results

### 3.1. Patients’ Demography

The participants’ demographic characteristics are shown in Table 1. A total of 392 patients were involved in this study, including 187 and 205 patients before and after the protocol updates, respectively. The mean age of the patients was higher in the after group than in the before group (57.9 ± 17.5 vs. 53.9 ± 19.4 years, respectively; *p* = 0.035). Most patients were admitted to the ICU for medical emergencies in both groups (84.0% vs. 80.0%; *p* = 0.310). Both groups had similar Acute Physiology and Chronic Health Evaluation scores (APACHE II scores) and Sequential Organ Failure Assessment (SOFA) scores (17.5 ± 7.1 vs. 17.8 ± 6.2, *p* = 0.600; 6.3 ± 3.6 vs. 6.5 ± 3.5, *p* = 0.612), but exhibited differences in the ratios of the arterial partial pressure of oxygen (PaO_2_) to the fraction of inspired oxygen (FiO_2_) (P/F) (223.2 ± 127.0 vs. 189.8 ± 116.6; *p* < 0.010). Chronic liver disease was more common in the before group than in the after group (4.3% vs. 12.2%; *p* = 0.005). Although there were differences in some demographic and clinical characteristics, we considered the incidence of VAP in the two groups to be comparable.

### 3.2. Bacterial Isolates from VAP

Both before and after the protocol updates, the main pathogen isolated from VAP was *Acinetobacter baumannii* (53.8% vs. 43.5%), followed by *Klebsiella pneumoniae* (23.1% vs. 21.7%), respectively (Table 1). Both were multidrug-resistant (MDR). There was no significant change in the causative bacteria of VAP before and after the protocol updates.

### 3.3. Compliance Rates for VAP Care Bundle

The VAP care bundle compliance rates before and after the protocol updates were 80.6% and 81.0%, respectively. The compliance rates for each item before and after the protocol updates were as follows: (i) 96.0% and 95.9% for hand hygiene, (ii) 95.4% and 95.6% for head elevation (gatch up 30–45°), (iii) 93.6% and 95.6% for oral care, (iv) 61.4% and 65.2% for oversedation avoidance, (v) 93.0% and 95.3% for breathing circuit management, (vi) 94.3% and 95.3% for cuff pressure control, (vii) 46.6% and 11.2% for subglottic suctioning of secretions, (viii) 68.1% and 78.0% for daily assessment for weaning and a spontaneous breath trial (SBT), (ix) 59.4% and 79.3% for early mobilization and rehabilitation, and (x) 96.1% and 99.9% for prophylaxis of peptic ulcers and deep-vein thrombosis (DVT).

The compliance rates were increased before and after the protocol updates for several items, such as weaning and SBT and early mobilization and rehabilitation; however, the compliance rate for subglottic suctioning of secretions decreased from 46.6% to 11.2%. There were no significant differences for other items.

### 3.4. VAP Incidence

Figure 1 shows the changes in VAP incidence over the study period. The VAP incidence rates per 1000 mechanical ventilation days were 27.0 and 11.3 before and after the protocol updates, respectively (*p* < 0.001; Table 1). The survival probability also improved (*p* = 0.049) (Figure 2b). Figure 2 shows graphs of the proportion of VAP patients–ICU days (Figure 2a) and the survival probability–ICU days (Figure 2b) before and after the protocol updates, respectively. The mean survival time (MST) was 24.0 and 45.0 days before and after, respectively. The lower VAP incidence rate and the longer survival probability in the after group show the effectiveness of the protocol updates with interactive education.

## 4. Discussion

This study provides unique insights into the implementation of a VAP care bundle. In general, compliance with a VAP care bundle is regarded as the main factor associated with VAP incidence reduction [19,20]; however, VAP incidence was not reduced despite > 80% compliance rates in this study. VAP incidence decreased from 27.0 to 11.3 per 1000 ventilator days (*p* < 0.001) after the protocol updates for the two items (‘oral care’ and ‘breathing circuit management’) and the head and chief nurses started the interactive communication and bedside education.

The ICU experts encountered difficulties when the VAP incidence was slightly and not significantly reduced during the first six months of the VAP care bundle’s implementation; although a review study reported that the higher the VAP care bundle compliance rate, the better the outcomes in general [33], VAP incidence did not show a significant decrease despite the care bundle compliance rate being >80% in this study. Some recent studies also reported no significant decrease in VAP incidence despite using a VAP care bundle [10,15,18,34]. A review study by Mastrogianni et al. [15] reported that 4 out of 38 studies indicated a low VAP reduction with a significant increase in VAP bundle compliance; however, the care bundle compliance rate was >80% in this study but did not significantly increase in this study. The care bundle compliance rate was not mentioned in other previous studies [10,18,34]. As one review study has reported that simply recommending a VAP care bundle is not enough [33], it is important to take steps if the VAP incidence does not decrease; for example, to make the VAP care bundle work in clinical practice as in this study, consider variations in patient demographics, such as the severity of the illness, which are not measured by ‘VAP rates per 1000 ventilator bed days’ [35], and so on.

In general, information is disseminated unidirectionally from experts to uneducated nonexperts. In high-technology environments, such as ICUs, conditions are dynamic, and information is frequently revised. This setting is similar to that of emergency departments and public health emergencies. In public health emergencies, the ‘interactive process of exchange of information and opinions between risk assessors, risk managers, and other interested parties’ is crucial [36]. However, before the updates, we disseminated the 10 items of the care bundle via posters and staff meetings, and we lacked interactive communication.

One reason why the procedure was highly variable between nurses was a lack of prioritization of optimal treatment practices amongst recommendations in resource-limited situations. The experts initially recommended cetylpyridinium chloride (CPC) for oral care, but it was unavailable in Vietnam [25]. Therefore, before the protocol was standardized, some nurses applied chlorhexidine, and others continued rinsing the mouth with tap water after brushing patients’ teeth with toothpaste, as they believed that toothpaste would be more comfortable for patients.

Hospital directors must be consistently engaged to facilitate the implementation of protocols. Multiple-use (reusable) breathing circuits were allowed because of the large number of different types of ventilators in use, and disposable breathing circuits are costly. However, after considering the financial balance in the general ICU with hospital managers, including the costs of treating VAP and extended ICU stays, the hospital management accepted the use of chlorhexidine and single-use breathing circuits as standardized VAP care bundle protocols. It is essential to optimize available resources to enhance staff competency [37] and involving hospital managers is a crucial factor for success in resource-limited situations.

In addition, while there are evidence-based VAP protocols for the VAP care bundle, nurses may interpret and apply them differently depending on how they comprehend the recommendations or whether certain protocols are unclear. Previous studies have found that barriers to VAP care bundle implementation were lack of skills, knowledge, equipment and supplies, education, and so on [38,39,40,41]. Previous studies revealed the importance of set-up and updates to protocols [38,42], and daily audits on compliance with a VAP bundle with weekly feedback [4,15,18,43,44]. During this study, the head nurses continued bedside training and coaching. They explained the gaps in care and necessary improvements to staff nurses. Nurses improved their clinical skills and motivation via this bedside daily training, as they observed the number of VAP patients reduced via their efforts.

Subglottic suctioning of secretions decreased from 46.6% before to 11.2% after the interventions. This happened because the suction endotracheal tube was supplied at the start of VAP bundled care; however, the supply gradually decreased. Hence, the staff could not suction secretions over the cuff, and they suctioned secretions continuously at the pharynx. Since alternative measures were taken even when suctioning over the cuff was not possible, it is unlikely that the lower compliance rate of the component strongly affected the VAP incidence rate over time.

This study had several limitations. First, quality improvement studies tend to be suitable for before-and-after studies. However, a before-and-after study cannot identify a causal relationship between an intervention and an outcome due to the lack of a control group. Second, our data represent a cohort from a single center, which might have a relatively small number of participants and uncontrolled confounding bias. Third, the self-report evaluation could be a potential bias; even the chief and head nurses re-evaluated after the staff nurse self-evaluated. Therefore, our results are suggestive but must be interpreted with caution because our interventions might not be generalizable. Although the 10-item VAP care bundle developed through protocol updates with interactive education successfully reduced VAP in Vietnam and Japan, this method may not be generalizable to other resource-limited settings. Further studies are necessary to explore whether protocol updates with interactive education could be potential strategies for addressing VAP prevention and other healthcare challenges in resource-limited settings.

## 5. Conclusions

The effectiveness of interventions through protocol updates with interactive education needs further study. Nevertheless, this intervention with a 10-item VAP care bundle, which initially slightly and not significantly decreased the VAP incidence rate, significantly decreased VAP incidence in the general ICU of a tertiary hospital in Vietnam. This intervention has the potential to improve the quality of care and make a VAP care bundle work in a resource-constrained and multidrug-resistant environment.

## Figures and Tables

**Figure 1 healthcare-13-00443-f001:**
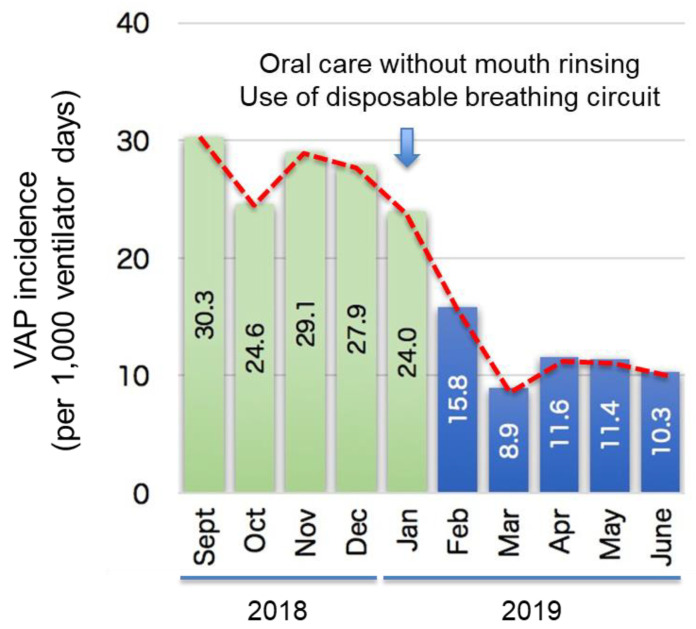
Change in VAP incidence over the observation period. VAP: ventilator-associated pneumonia.

**Figure 2 healthcare-13-00443-f002:**
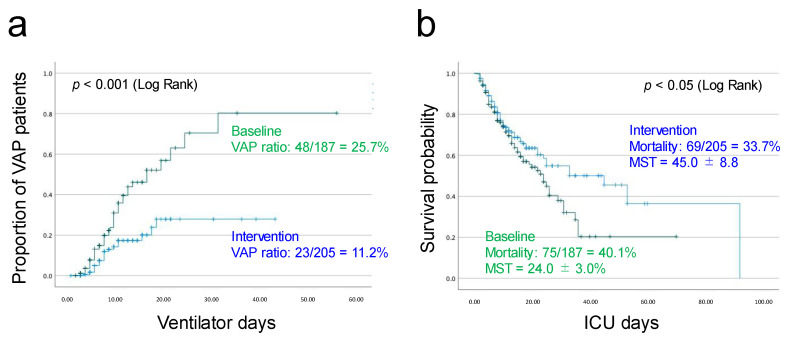
Graphs of proportion of VAP patients–ICU days (**a**) and survival probability–ICU days (**b**) before and after updates of the VAP care bundle (oral care and breath circuit management) and interactive communication and bedside education. VAP: ventilator-associated pneumonia; ICU: intensive care unit; MST: median survival time.

**Table 1 healthcare-13-00443-t001:** Demographic and clinical characteristics of ventilated patients before and after protocol updates of ‘oral care’ and ‘breathing circuit management’ among the 10-item VAP care bundle and interactive communication and bedside education in the general intensive care unit at a tertiary hospital in Vietnam from September 2018 to June 2019.

Variable	All	Before	After	*p*-Value
		(Sep 2018 to Jan 2019)	(Feb 2019 to Jun 2019)	
Patients, n	392	187	205	--
Male gender, n (%) *	223 (59.3)	103 (56.9)	120 (61.5)	0.401
Age, mean ± SD *	56.0 ± 18.6	53.9 ± 19.4	57.9 ± 17.5	**0.035**
APACHE II score, mean ± SD **	17.6 ± 6.7	17.5 ± 7.1	17.8 ± 6.2	0.600
SOFA score, mean ± SD **	6.4 ± 3.5	6.3 ± 3.6	6.5 ± 3.5	0.612
Laboratory test (mean ± SD)				
Albumin (mg/mL) **	29.6 ± 18.3	30.3 ± 25.3	29.0 ± 6.7	0.501
PaO_2_/FiO_2_ ratio **	206.1 ± 122.7	223.2 ± 127.0	189.8 ± 116.6	**0.010**
Platelet (×10^4^/µL) **	205.2 ± 133.4	207.4 ± 135.6	203.1 ± 131.5	0.762
Bilirubin (µmol/L) **	30.8 ± 61.5	29.4 ± 64.2	32.2 ± 59.0	0.664
Creatinine (µmol/L) **	163.8 ± 179.4	169.4 ± 191.3	158.4 ± 167.7	0.563
Underlying disease, n (%)				
Chronic respiratory disease	42 (10.7)	22 (11.8)	20 (9.8)	0.521
Chronic cardiovascular disease	118 (30.1)	54 (28.9)	64 (31.2)	0.614
Chronic liver disease	33 (8.4)	8 (4.3)	25 (12.2)	**0.005**
Chronic kidney disease	34 (8.7)	17 (9.1)	17 (8.3)	0.779
Diabetes	53 (13.5)	19 (10.2)	34 (16.6)	0.063
Immunocompromised state	50 (12.8)	28 (15.0)	22 (10.7)	0.209
Septic shock	17 (4.3)	5 (2.7)	12 (5.9)	0.123
Cancer	17 (4.3)	11 (5.9)	6 (2.9)	0.151
Type of admission, n (%)				
Scheduled surgery	11 (2.8)	6 (3.2)	5 (2.4)	0.645
Emergency surgery	17 (4.3)	7 (3.7)	10 (4.9)	0.582
Coronary intervention	11 (2.8)	4 (2.1)	7 (3.4)	0.445
Medical emergency	321 (81.9)	157 (84.0)	164 (80.0)	0.310
Trauma	0 (0.0)	0 (0.0)	0 (0.0)	n/a
Outcomes				
ICU length of stay (days), mean ± SD **	14.1 ± 11.9	12.9 ± 10.1	15.2 ± 13.3	0.054
Ventilator days (days), mean ± SD **	9.7 ± 8.9	9.5 ± 8.7	9.9 ± 9.0	0.636
VAP, n (%)	71 (18.1)	48 (25.7)	23 (11.2)	**<0.001**
VAP incidence rate (per 1000 MV days)	18.7	27.0	11.3	**<0.001**
Bacterial isolate(s) from VAP patients ***				
*Acinetobacter baumannii*	31 (50.0)	21 (53.8)	10 (43.5)	
*Klebsiella pneumoniae*	14 (22.6)	9 (23.1)	5 (21.7)	
*Pseudomonas aeruginosa*	3 (4.8)	3 (7.7)	0 (0.0)	
*Acinetobacter baumannii* and *Klebsiella pneumoniae*	2 (3.2)	0 (0.0)	2 (8.7)	
Others	12 (19.4)	6 (15.4)	6 (26.1)	
Outcome (%)				
Death by VAP	17 (4.3)	13 (7.0)	4 (2.0)	
Death by primary disease	127 (32.4)	62 (33.2)	65 (31.7)	
ICU leave	248 (63.3)	112 (59.9)	136 (66.3)	

* 16 data points were missing; ** 31 data points were missing; *** 9 data points were missing. ICU length of stay—the number of days from admission to discharge or death of the patient; n/a—not available. Bold indicates statistically significant as *p* < 0.05.

## Data Availability

The data set used in this study is available from the corresponding author on reasonable request.

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
