# Peer review of "The Development of a 10-Item Ventilator-Associated Pneumonia Care Bundle in the General Intensive Care Unit of a Tertiary Hospital in Vietnam: Lessons Learned"

_healthcare, 2025, doi:10.3390/healthcare13050443_

Round 1

Reviewer 1 Report

Comments and Suggestions for Authors

I appreciate the opportunity to review the manuscript entitled "Development of a 10-item ventilator-associated pneumonia (VAP) care bundle in a general intensive care unit of a tertiary hospital in Vietnam: lessons learned".

In general, the article addressed a highly relevant and timely topic related to a breakthrough in the high incidence of VAP through bundled care.

However, I would like to offer a few suggestions that I believe could enhance the clarity and impact of the manuscript.

The introduction presents relevant information; however, the transition between paragraphs could be improved. The integration of transition phrases to facilitate the flow of ideas, particularly in the context of the discussion on the significance of prevention and the proposed interventions, would enhance the coherence of the manuscript.

The objective of the study could be more clearly outlined at the conclusion of the introduction. In addition, the inclusion of a sentence that specifically articulates the research questions and hypothesis would serve to more definitively delineate the study's focal point.

It is imperative to verify that all cited references are current and pertinent to the subject matter. Some may be more recent or more pertinent to the current context of VAP (Guillamet CV, Kollef MH. Is Zero Ventilator-Associated Pneumonia Achievable? Updated Practical Approaches to Ventilator-Associated Pneumonia Prevention. Infect Dis Clin North Am. 2024 Mar;38(1):65-86. doi: 10.1016/j.idc.2023.11.001; Sabrah NYA, Pellegrino JL, Mansour HE, Mostafa MF, Kandeel NA. Care Bundle Approach for Oral Health Maintenance and Reduction of Ventilator-Associated Pneumonia. Crit Care Nurs Q. 2024 Oct-Dec 01;47(4):335-345. doi: 10.1097/CNQ.0000000000000522.; Rosenthal VD, Jin Z, Yin R, Sahu S, Rajhans P, Kharbanda M, Nair PK, Mishra SB, Chawla R, Arjun R, Sandhu K, Rodrigues C, Dongol R, Myatra SN, Mohd-Basri MN, Chian-Wern T, Bhakta A, Bat-Erdene I, Acharya SP, Alvarez GA, Moreno LAA, Gomez K, da Jimenez-Alvarez LF, Henao-Rodas CM, Valderrama-Beltran SL, Zuniga-Chavarria MA, Aguirre-Avalos G, Hernandez-Chena BE, Sassoe-Gonzalez A, Aleman-Bocanegra MC, Villegas-Mota MI, De Moros DA, Castaneda-Sabogal A, Carreazo NY, Alkhawaja S, Agha HM, El-Kholy A, Abdellatif-Daboor M, Dursun O, Okulu E, Havan M, Yildizdas D, Deniz SSO, Guclu E, Hlinkova S, Ikram A, Tao L, Omar AA, Elahi N, Memish ZA, Petrov MM, Raka L, Janc J, Horhat-Florin G, Medeiros EA, Salgado E, Dueñas L, Coloma M, Perez V, Brown EC. Assessing the impact of a multidimensional approach and an 8-component bundle in reducing incidences of ventilator-associated pneumonia across 35 countries in Latin America, Asia, the Middle East, and Eastern Europe. J Crit Care. 2024 Apr;80:154500. doi: 10.1016/j.jcrc.2023.154500.)

The methods section could include more details about the patient selection process and data collection. For instance, what methodology was employed to select the study's participants? What exclusion criteria were applied?

The care description of the VAP bundle could be more detailed, with explanations of the evidence supporting each item in the bundle.

While the statistical analysis is described, providing a concise rationale for the selection of statistical tests would be beneficial, particularly for readers lacking familiarity with statistical analysis.

The results are presented in a clear and coherent manner. However, the section could benefit from the addition of subsections to enhance its readability. For instance, the inclusion of subsections that address demographics, VAP incidence rates, and compliance rates would further facilitate comprehension.

Rather than merely presenting the data, it would be beneficial to incorporate a concise interpretation of the results as they are presented. While tables and figures are beneficial, it is crucial to ensure that all tables are accompanied by clear captions and that the data is explained in a manner that is accessible in the text.

The discussion could benefit from a more robust connection to existing literature. Making comparisons with previous studies on VAP and care packages would be valuable to contextualize the findings.

While acknowledging the limitations, a more thorough discussion is necessary to elucidate how these limitations may affect the interpretation of the results and the applicability of the conclusions.

The practical implications of the findings could be better elaborated, especially in a context of limited resources.

The conclusions section should summarize not only the results, but also the implications for clinical practice and future research.

The article's formatting and stylistic choices are not optimal.

The article contains grammatical and typographical errors that must be corrected. A meticulous review is imperative to ascertain the clarity and coherence of the text.

A uniform terminology should be employed consistently throughout the article. For instance, the utilization of "VAP care bundle" and "VAP care package" should be uniform.

Reviewer 2 Report

Comments and Suggestions for Authors

The manuscript address the question of how implementing and updating a 10-item ventilator-associated pneumonia (VAP) care bundle can reduce VAP incidence in a resource-limited environment. Overall, this manuscript is suitable for publication in this journal. However, there are some points that need to be addressed. I will present these for each part of the manuscript as follows:

I am very pleased with the title, as it accurately reflects the study's focus on developing and implementing a 10-item ventilator-associated pneumonia (VAP) care bundle in a general intensive care unit in Vietnam "specificty of geographic location", as well as the lessons learned from the process.

The abstract provides a clear and concise summary of the study, including the background, objectives, methods, results, and conclusions. It effectively highlights the key findings, such as the reduction in VAP incidence following protocol updates and interactive education. However, some details could be added to enrich it. For example, consider specifying key results with numerical data to provide a more robust summary. Additionally, ensure that all keywords are indexed terms commonly used in this field to improve discoverability, as mismatched keywords may affect the article's visibility on platforms like PubMed.

The Introduction provides sufficient background on ventilator-associated pneumonia (VAP), its global burden, and its particular challenges in resource-limited settings. I am very supportive of this, as it ensures a broad understanding of the context, even for researchers outside the field. However, some technical terms, such as 'care bundle,' require brief explanations to enhance accessibility. Additionally, the Introduction clearly identifies gaps in the effectiveness of care bundles and highlights the need for tailored approaches in resource-constrained environments. However, the justification for the study should be strengthened by emphasizing the novelty of combining protocol updates with interactive education.

While the references are relevant, there is room to include studies on similar interventions in other low-resource settings for comparison and key guidelines or systematic reviews on VAP prevention strategies.

The aims of the study are clearly stated and align well with the research methodology and conclusions. However, they need to be explicitly connected to the expected outcomes, such as reducing VAP incidence and improving compliance rates. I would prefer to see outcome-based aims to strengthen the focus and clarity of the study.

I am pleased that this study specifies the location (ICU at Bach Mai Hospital), time frame (September 2018 to June 2019), and inclusion criteria for patients. The study design and methods are clearly described, including the protocol updates and educational interventions. However, it would be beneficial to explain the rationale for choosing a before-and-after design and to acknowledge its inherent limitations. While the Methods section provides a clear description of how data were collected, including compliance rates, bacterial cultures, and VAP incidence, more detailed explanations of certain procedures (e.g., how compliance was assessed and recorded) would enhance clarity. Additionally, specific details on the data collection process, such as timelines, frequency of monitoring, and roles of staff involved in assessments, should be included to improve transparency.

The aouthors does not  specify exclusion criteria for the study. This is an important omission, as exclusion criteria are crucial for defining the study population and ensuring the validity of the results.

The primary outcome (VAP incidence) and secondary outcomes (compliance rates and bacterial isolates) are clearly defined. However, more details are needed on the methods for diagnosing VAP and calculating compliance rates. I suggest providing specific diagnostic criteria for VAP, referencing validated guidelines, and explaining how compliance was measured and objectively validated.

While the authors mention tools such as compliance records, they do not provide detailed descriptions or cite relevant published studies, if applicable.  If a validated questionnaire or tool was used, include a summary and proper citation in the text. Additionally, describe how these tools were adapted or applied in the context of this study.

The sample size (392 patients) seems appropriate. However, a justification for the sample size should be provided.

The manuscript does not explicitly state whether the data met the assumptions for the statistical tests. It is important to confirm whether the data were tested for normality before using t-tests. If the data did not meet these assumptions, describe any  alternative methods that were applied. Also, you need to clarify the statistical methods used, especially the Kaplan–Meier analysis.

The results are central to the study’s aims, such as the reduction in VAP incidence and changes in compliance rates. The table headings and figure legends are informative and sufficiently detailed. However, for each figure, the phrase 'Figure 1' or 'Figure 2' appearing inside the figure itself should be omitted, as the legends already provide very good and informative descriptions. If the data are available, include more detailed compliance trends for individual components of the VAP care bundle over time. Additionally, add more granular information about how different factors (e.g., patient demographics) influenced outcomes. Importantly, ensure the Results section remains focused on factual statements about the data. Move any interpretations (e.g., explanations for why compliance rates for subglottic suctioning decreased) to the Discussion section.

The Discussion aligns well with the aims of the study. However, make sure that every point in the Discussion explicitly ties back to the study’s stated aims and objectives.

The authors compare their findings with previous studies, but the comparisons are somewhat brief and could be expanded. I suggest including more specific comparisons with studies conducted in similar resource-limited settings. Additionally, discuss how the findings relate to global VAP prevention efforts, highlighting both similarities and differences.

The authors briefly mention how their findings could inform VAP prevention strategies in resource-limited settings but do not elaborate on specific future research directions. I suggest identifying areas for further research and discussing the potential for adapting the interactive education model to address other healthcare challenges.

The authors acknowledge some limitations, such as the single-center study design and lack of control groups. However, additional limitations, such as potential biases in self-reported compliance data and limited generalizability, are not addressed.

I am very pleased and stastified that the conclusions are supported by the data and are appropriately cautious. The authors effectively emphasize the importance of interactive education and protocol updates without overstating their impact.

The references cited in this manuscript appropriate and relevant to this research.

Reviewer 3 Report

Comments and Suggestions for Authors

The article  addresses very common but important issue in the intensive care management.  VAP has been extensively studied and thorough guidelines and instructions exist. The recommendations in these guidelines may not be suitable for low resource health care systems. The experimental improvisation implemented by authors may not appear innovative but may suggest important modifications in similar settings.

The decrease in VAP incidence after  updating of  the two items protocol and interactive communication and education in authors institute can be understood as quality improvement activity rather than a research  breakthrough.  

The manuscript is well organised and meticulously presented. An overview ( e.g. in the form of a table ) of compliance with the 10 items before and after updating may further enhance the comprehension. 

Round 2

Reviewer 1 Report

Comments and Suggestions for Authors

Following meticulous review of the reviews and the authors' responses to my prior recommendations, it is evident that the authors have adequately addressed all of my recommendations. The improvements implemented in the manuscript have resulted in a clearer and more coherent text, with a more robust presentation of the data and a more in-depth discussion of the existing literature.

Noteworthy enhancements include the following:

The introduction now features smoother transitions between paragraphs, facilitating readers' understanding of the context and importance of the study. The study's objective and research questions are articulated with greater clarity, offering a more focused overview. The methods section has been expanded to include details on the patient selection process and exclusion criteria. The descriptions of the VAP care bundle items have been enriched with evidence supporting each component. The results section has been meticulously organized into subsections, thereby facilitating the presentation of the data and enhancing its interpretability. The discussion has been expanded to establish stronger connections with the existing literature, better contextualizing the study findings. Additionally, the authors have meticulously revised the text, correcting grammatical and typographical errors, and ensuring uniformity in the terminology used.

In light of the substantial improvements made, I recommend the publication of this manuscript in the journal Healthcare. The research presented here contributes significantly to the field and has the potential to benefit both clinical practice and future research.

Reviewer 2 Report

Comments and Suggestions for Authors

The manuscript looks improved. Good luck!